# A simple, scalable approach to building a cross-platform transcriptome atlas

**Paul W. Angel**[1], **Nadia Rajab**[1], **Yidi Deng**[2], **Chris M. Pacheco**[1], **Tyrone Chen**[1], **Kim-Anh Lê Cao**[2], **Jarny Choi**[1☯], **Christine A. Wells**[1☯]*

**1** Centre for Stem Cell Systems, The University of Melbourne, Melbourne, Victoria, Australia, **2** Melbourne Integrative Genomics, School of Mathematics and Statistics, The University of Melbourne, Melbourne, Victoria, Australia

☯ These authors contributed equally to this work.

* wells.c@unimelb.edu.au

**Data Availability Statement:** No new data has been generated as part of this study, but primary data has been taken from public repositories, NCBI

## Abstract

Gene expression atlases have transformed our understanding of the development, composition and function of human tissues. New technologies promise improved cellular or molecular resolution, and have led to the identification of new cell types, or better defined cell states. But as new technologies emerge, information derived on old platforms becomes obsolete. We demonstrate that it is possible to combine a large number of different profiling experiments summarised from dozens of laboratories and representing hundreds of donors, to create an integrated molecular map of human tissue. As an example, we combine 850 samples from 38 platforms to build an integrated atlas of human blood cells. We achieve robust and unbiased cell type clustering using a variance partitioning method, selecting genes with low platform bias relative to biological variation. Other than an initial rescaling, no other transformation to the primary data is applied through batch correction or renormalisation. Additional data, including single-cell datasets, can be projected for comparison, classification and annotation. The resulting atlas provides a multi-scaled approach to visualise and analyse the relationships between sets of genes and blood cell lineages, including the maturation and activation of leukocytes in vivo and in vitro.

In allowing for data integration across hundreds of studies, we address a key reproducibility challenge which is faced by any new technology. This allows us to draw on the deep phenotypes and functional annotations that accompany traditional profiling methods, and provide important context to the high cellular resolution of single cell profiling. Here, we have implemented the blood atlas in the open access Stemformatics.org platform, drawing on its extensive collection of curated transcriptome data. The method is simple, scalable and amenable for rapid deployment in other biological systems or computational workflows.

## Author summary

Combining data from many different studies is an attractive way of capturing new aspects of the biology being studied. Biological variance attributable to cell type, cellular niche, origin, disease status or environmental stimuli is the basis of most small-n transcriptome

GEO; EMBL-EBI ArrayExpress; and the Human Cell Atlas portal and accessions are listed in the manuscript, S1 Table. All processed files used to construct the atlas are available from the Stemformatics database, where links to the original publications and data are provided.

**Funding:** Stemformatics was established through Australian Research Council Funding to Stem Cells Australia (SRI110001002)and to CAW (Future Fellowship FT150100330) (https://www.arc.gov.au). KALC was supported by the National Health and Medical Research Council (NHMRC) Career Development fellowship (GNT1159458). PWA and JC are funded by NHMRC (GNT1181327) and (APP1186371) to CAW (https://www.nhmrc.gov.au). NR is funded by the Centre for Stem Cell Systems (https://biomedicalsciences.unimelb.edu.au/departments/anatomy-and-neuroscience/engage/cscs) and the CSIRO Synthetic Biology Future Science Platform (https://research.csiro.au/synthetic-biology-fsp/). The funders had no role in study design, data collection and analysis, decision to publish, or preparation of the manuscript.

**Competing interests:** The authors have declared that no competing interests exist.

studies. In aggregation, these promise to capture emergent dimensions of a biology that is not possible to view from any individual study. However biological signal is easily swamped by technical artifact, especially when data is generated on platforms with profoundly different data structures. This is the case when comparing microarray data to RNAseq, or RNAseq to single cell profiling. Consequently, transcriptome atlases are generally comprised from a small number of donors/conditions surveyed using one technology platform.

In this paper we present a simple and scalable data integration method that is platform agnostic. We provide a proof-of-principle by constructing an atlas of blood cells that combines many data sets measured on different platforms, and that in combination, recapitulates the known blood hierarchy. The atlas provides a reference to compare external samples to, allowing users to benchmark new derivation or isolation methods. It also provides a reference point for new data types, such as the classification of single cells. The approach allows for FAIR data reuse and robust identification of molecular signatures across multiple studies and experimental conditions.

## Introduction

RNA profiling has been a mainstay descriptor of cellular systems for over two decades, but methods for measuring transcript abundance have changed dramatically over this period. The field was revolutionised by microarrays, which allowed simultaneous hybridisation and colourmetric read out for a catalogue of known genes [1]. Microarrays were rapidly adopted because they were a fast, inexpensive and simple way to measure the transcriptional output of a biological system. However, the need to predefine sequences to be interrogated, and a linear range constrained by the stoichiometry of probe and target meant that microarray platforms were rapidly superseded by RNA sequencing (RNAseq) technologies. Now the most prevalent experimental platform, the range of detected transcripts is determined by the number of tags counted in a sequencing run and cellular resolution is determined by the complexity of the profiled population [2, 3]. Increased resolution has escalated rapidly with the advent of single-cell RNA sequencing (scRNAseq) technologies [4, 5]. Although some platforms are being refactored and repurposed, such as the reinvention of hybridisation-based platforms for spatial profiling [6], successive technologies become rapidly redundant, as does the data generated on them.

There is a need to move past information gathering and to move towards build new knowledge frameworks. Yet technological change drives much recursive data derivation. This represents a massive intellectual and financial investment by research groups and funders on data that is not adequately being reused, despite its availability in data repositories [7, 8]. A wealth of information still resides in data generated on obsolete technologies: these collectively represent a large back catalogue of carefully phenotyped cells and meticulous experimental systems that can be viewed one study at a time in platforms such as ArrayExpress [9]. A major barrier to data reuse is the computational capacity to directly integrate and compare successive technologies. While the drivers for platforms are increased sensitivity and resolution of systems-scale measurements, it remains difficult to benchmark the new against the old.

Drawing on curated knowledge commons is particularly important when new platforms, such as scRNAseq, rely on annotations from post-hoc analysis rather than starting with well phenotyped cells. The methods that are most commonly used to integrate scRNAseq with different platforms rely on projection or harmonisation of different data types onto a reference

scRNAseq data set, and are designed to compare data in a pairwise manner, so are not easily scaled to include many experimental series [10]. In order to take advantage of the back-catalogue of phenotype-gene expression data, we need new approaches to combine experimental series from several different platforms and across multiple studies.

Combining RNAseq with the microarray is particularly challenging because data are acquired in a continuous (microarray) or discrete (RNAseq) manner, and the number of genes captured in a single cell may be orders of magnitude less than that measured in a population. While it is most common to combine data from the same microarray platform (e.g. [11, 12]) or RNAseq (e.g. [13, 14].) combining different types of platforms is less common (e.g. see also [15]). Combining microarrays with RNAseq has been previously attempted [16, 17], however, these methods focus on global normalisation, which has a major impact on stability and scalability when new data is imported. Many normalisation approaches that account for platform variance require prior identification of sample groups that are expected to harmonise together. This can introduce class biases, whilst also enforcing such strong transformations to data structure that meaningful biological signal is removed—these are acknowledged problems with batch correction methods such as COMBAT ([18]), and RUV-III ([19]). Class imbalance is typically encountered when attempting to merge a small number of data sets. For example, when benchmarking a new sample type against an existing exemplar the lack of common or appropriate reference samples in the comparison, as well as prior designation of sample class in the normalisation structure can lead to spurious claims of cell-type similarity. This could be addressed if new data could be compared to a reference atlas series, but no such benchmark exists.

Here, we use the Stemformatics catalog [20], which has curated hundreds of studies, to assess the extent that platform impacts on expression variance for each gene. This challenged our initial assumption that accounting for batch necessitates an adjustment to every gene expression value. We selected a subset of genes with low attributable platform variance to compile samples from many studies, resulting in a reference atlas that reflects cell properties that are independent of mode of measurement. By including sufficient representation across different cell types we gain insights into the behaviour of related cell types, whilst also providing a platform for further analysis (e.g. comparisons between disease and normal states, or between in vitro and in vivo models); and to benchmark new platforms, including scRNAseq.

## Materials and methods

In designing this method, the effect of platform is assumed to be systematic variation, and other batch effects will be averaged out by the multiple datasets covering the biology. We test these assumptions by leveraging the large collection of data in Stemformatics which samples different platforms and numerous cell types. The method introduced here assesses each gene independently to quantify the impact of experimental platform on that gene's expression across the whole data series. Genes with low platform effect are selected for subsequent analyses.

### Data curation

All data used to compile the blood atlas was curated for data quality, and for method of cell isolation and phenotyping. This metadata is captured in the Stemformatics annotation table (available to download at https://www.stemformatics.org/atlas/blood), and includes tissue source, antibody profiles where bead or FACs isolation is used, age of donor (fetal, neonatal, adult). Cells that are profiled directly from tissue are annotated as an in vivo source; mature cells isolated from blood or bone marrow and cultured for any period of time are labelled ex

vivo; and cells differentiated in the laboratory from hematopoietic progenitor (typically mobilised peripheral blood, bone marrow or cord blood) or from a pluripotent cell source are labelled in vitro. This information is available to the viewer in the Stemformatics implementation of the blood atlas; primary data sources and publications are linked from every data set page. Supplementary S1 Table summarises the datasets used to construct the atlas, along with external datasets used to project onto the atlas.

Not all samples were used from each data set to construct the atlas, which excluded cells from blood pathologies such as leukemic cell types. Note also that in early iterations of the atlas, T-cell subsets isolated using negative selection alone were found to have a high monocyte contamination when compared to T-cells isolated using flow cytometry gates, as evaluated by high expression of myeloid marker gene profiles CD14, CD16 and HLADR. Therefore samples isolated using negative-selection methods were excluded from the atlas unless further purification and phenotyping was provided by the authors.

The standard Stemformatics processing pipelines were implemented, where data was assessed for linear range/library size, RNA species, and RNA degradation using 5'/3' signals where appropriate for the profiling method. Datasets were excluded if they showed evidence of over-amplification, incomplete data availability in the public databases GEO or ArrayExpress, incomplete sample metadata or identified sample-swaps, or where experimental design was confounded. Details of the Stemformatics data curation pipeline are available [20, 21].

## Data transformation

Combining datasets measured on microarray platforms and RNAseq presents two main difficulties. Firstly, each platform produces data on a different scale, i.e. they measure abundance in different units. Secondly, microarrays are composed of gene probes, which are physically different and may be in principle measure transcripts not represented by alternate array models. These problems are addressed in two stages presented below, a data transformation stage and a gene filtering stage.

Only genes measurable in all of the available platforms are used to construct the atlas. In this instance we start with 13,661 genes. Expression values from RNAseq (RPKM) or microarray are transformed to the same scale. Microarrays have a component of lowly expressed genes at a non-zero value, whereas lowly expressed genes within RNAseq data can be exactly zero. Thus data structure (discrete vs. continuous) and sensitivity are quite different. Gene expression for each sample is transformed into rank percentile values—the gene with the highest expression is assigned a value of 1 and the lowest receives a value of 0. Values in between are uniformly spaced accorded to the rank of the genes expression. Tied values are given the same rank, which is average of their would-be ranks if they were not tied. Note that this scheme is scalable because the inclusion of new samples only requires that they are given the same ranked transformation, avoiding the need to continually renormalise the entire data series. The analysis of the influence of the rank transformation on the platform effect can be found in the S1 Text section 1.3.

## Variance modelling and gene selection

Principal component analysis (PCA) is performed to collate samples after the percentile transformation, in order to find reliable global structure [22]. As in Fig 1 middle row, there is a clear platform effect in the clustering of the samples, which must be suppressed. We estimate the platform effect on each gene by fitting a univariate linear model with platform as an

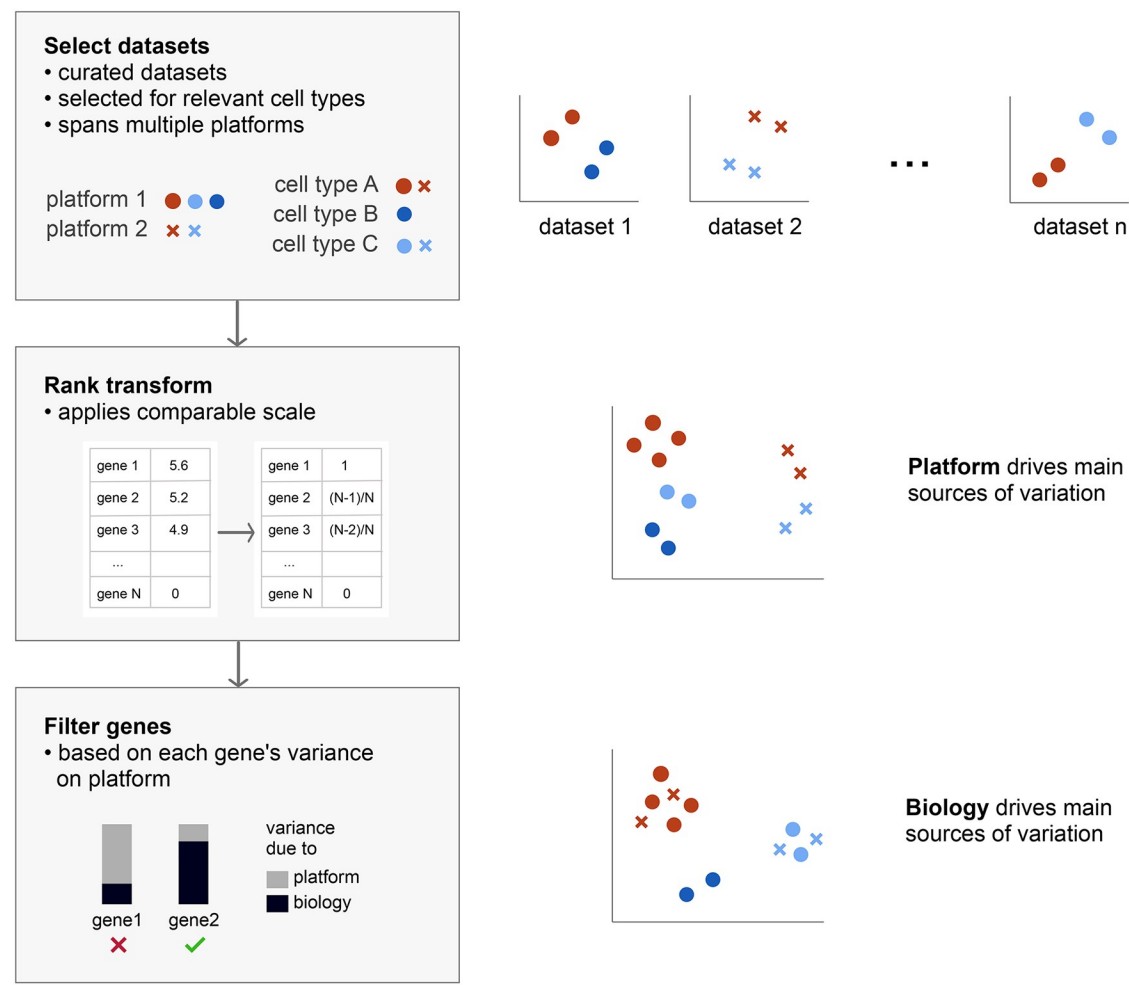

**Fig 1. The blood atlas is constructed by integrating many independent curated datasets.** Top row: the individuals PCAs of a set of quality-controlled independet datasets. These datasets are measured on a different platforms. Middle row: genes are rank transformed in order to move the expression distributions from the different platforms onto the same distribution. However, after running a PCA on the transformed data a platform clustering is still present. Bottom row: genes are univariately assessed for platform dependence, and filtered in order to keep only genes with a low fraction of the variance dependent upon platform. The resulting PCA then shows clustering based biological features.

independent variable,

$$y = X_p \beta_p + \epsilon, \qquad \epsilon \sim N(0, \sigma_\epsilon^2)$$

where $y$ is the expression of a single gene across all samples, $X_p$ indicates membership of the platform with coeffecient $\beta_p$. The variance attributable to platform is defined as

$$\sigma_p^2 = var(X_p \beta_p),$$

and the total variance

$$\sigma_{\text{Total}}^2 = \sigma_p^2 + \sigma_\epsilon^2.$$

therefore, the proportion of variance attributable to platform is

$$\frac{\sigma_P^2}{\sigma_{\text{Total}}^2}.$$

In practice this is implemented this using the variance partitioning package [23], with a single fixed effect (platform). This model is a fixed effect analysis of variation (ANOVA).

The distribution of variance attributable to platform is shown in Supplementary S1 Fig. Approximately 25% of genes examined were seriously impacted by platform, that is more than half of their variance was attributable to platform. Most genes were not overwhelmed by their method of measurement. In order to select the genes with minimal dependence upon platform a threshold of 0.2 of the variance of a gene is required. The PCA was constructed from this gene subset. The resulting PCA is shown in Fig 2 and effectively removes platform dependence. The process reducing platform dependence when lowering the threshold is outlined in S1 Text section S1.1 and S2 Fig. All PCA generation was implemented via the python scikit-learn package [24].

## Comparison of filtered and non filtered genes based on variance partitioning

To assess the effect of gene filtering in our approach, we partitioned the variance of all genes from the original data set (13661 genes) and calculated the variance explained by the 'Class' (sample source, progenitor type or cell type) and Platform using a linear mixed model (LMM) as follows.

Ranks percentiles were transformed using the probit function to fit the normality assumption of a LMM. For each gene $i$, $i = 1, \ldots, 13661$, we fitted a linear mixed model of the form

$$y_i \; = \mu \; + \; Z^{(1|Class)}\alpha_{Class} \; + \; Z^{(1|Platform)}\alpha_{Platform} \; + \varepsilon$$

with variance components

$$\alpha_{Class} \; \sim \; N\big(0, \; I\sigma_{class}^2\big)$$

and

$$\alpha_{Platform} \; \sim \; N\big(0, \; I\sigma_{Platform}^2\big).$$

The proportion of variance explained by class effect and platform effect was evaluated with the variancePartition R package [25] and genes were ordered according to their estimated class/platform variance ratio $\frac{\hat{\sigma}_{class}^2}{\hat{\sigma}_{Platform}^2}$.

## Clustering

In order to define regions of cohesive biology and test the stability of the atlas, we applied K-Means clustering to the principal components, which represent the coordinates of each sample in the 3D space. It is implemented via the sci-kit learn packages [24]. To provide a comparison, agglomerative (bottom up hierarchical) clustering is also implemented via the sci-kit learn package. Euclidean distance and Ward linkage was used in the Agglomerative algorithm. The two re-sampling schemes used were a jackknife re-sampling (leave-one-data-set-out), and bootstrap re-sampling performed 500 times.

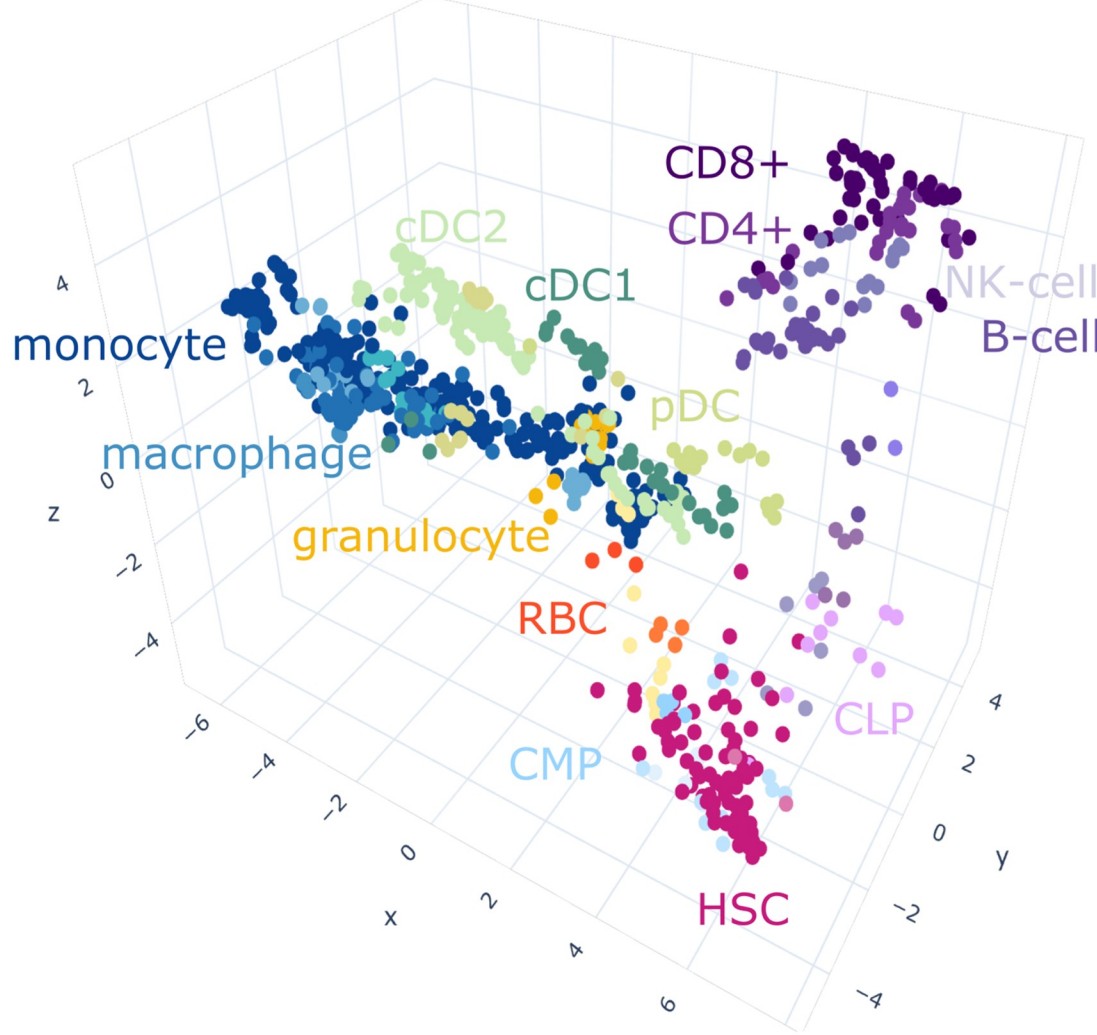

**Fig 2. The S4M blood atlas.** The S4M blood atlas integrates samples from 38 independent datasets. Each point is sample, and there are 3700 genes used in construction of the PCA. The colour indicates the annotated cell type. Progenitors sit in a region in the corner, while the the myeloid and lymphocyte arms separate out. The lymphocyte region includes both T and B cells. Dendritic cells sit in the cloud in the center if derived from an in vivo source, or cord-blood derived DC sit in a group.

Both cluster algorithms require the number of clusters, *k*, as an input parameter. Multiple values of *k* are assessed via a stability analysis based on re-sampling (described in S1 Text section S1.2), and the optimal *k* value was chosen as soon as the stability measure started to decrease. The stability measured used is the H-index, also outlined in S1 Text section S1.2.

## Projection of external data

To project new data sets onto the atlas, we transform the data as previously described into percentile values. Only genes selected in the construction of the original atlas are retained. The original PCA defines the graph coordinates system defined by principal components. Each

component is defined by a linear combination of genes, with each gene receiving a weight, also known as it's loading. Applying these coefficients to new data produces a coordinate in the PCA space for projection. The PCA and transformation is done with the scikit-learn [24].

If genes are missing from the projection data, they are given the lowest rank. These missing genes often result from slightly different genome annotations: microarrays particularly suffer from outdated probe annotations resulting in absent or misrepresentation of genes used to construct the atlas. If a large proportion of genes are missing, this will distort the projection, thus it is advisable to use caution when applying old or uncommon microarray platforms built on outdated genome versions. Note that Stemformatics workflows include alignment of microarray probes to the current genome version for gene annotation purposes.

A vignette is provided on the Stemformatics.org atlas website to assist users to project their own data to the atlas. This includes a detailed guide of file formats, and some recommendations for single cell projections.

## Single cell RNAseq expression data

Single cell RNAseq expression data was sourced from [26]. A pseudo-bulk aggregation method was used to aggregate cells belonging to the same cluster, and where the cluster identity was taken from the original publication [26]. In general, 8-10 cells were pooled per aggregate. We have previously shown that this number of cells in a 10X experiment allows for reasonable approximation of the data structure of the atlas data in [27]. Each cluster was randomly divided into subgroups such that each projected 'sample' had the same number of cells within it, and transcript reads from these cells were pooled to create a single pseudo-bulk sample for that subgroup. The subgroup has the same identity as the original group, so might be expected to project into the same region of the atlas. These pseudo-bulk samples were projected onto the atlas in the same manner as described above.

## Implementation and code availability

The Blood atlas and accompanying myeloid subset are available as interactive plots at www.stemformatics.org/atlas/blood and www.stemformatics.org/atlas/myeloid. These pages contain a number of features to help users navigate the atlas and perform useful functions:

- Interactive PCA with 3d/2d toggle.

- Colour by sample group, such as progenitor type or cell type.

- Show gene expression profile as a colour gradient.

- View gene expression and colour by sample group side by side.

- Project RNA-Seq dataset hosted at Stemformatics after a search.

- Project one's own dataset by providing expression and sample files as text files.

- Show and find which samples from which datasets make up the atlas.

- Download relevant data files (rank transformed expression and annotation tables) used by the atlas.

- Download plot in custom size.

Python code which can be used to manipulate the atlas data, to recreate the PCA for example, is available at https://bitbucket.org/stemformatics/s4m_pyramid/src/master/scripts/atlas.py.

## Results

### Recursive application and unsupervised K-means clustering upon the blood data

The method relies on several assumptions: (1) the biology of interest can be represented by the expression of many genes, (2) across platforms, some genes are measured less consistently than others, but there is a subset of genes where platform contributes substantially less to gene expression variability than the biology of interest, and therefore (3) the biology of a cell can be meaningfully described at several scales by identifying subsets of molecular attributes that are selected on cross-platform performance.

The method in its simplest implementation is agnostic to the presence of a biological signal or other confounding technical variables, but these can be subsequently applied to assess the major sample groupings. Here, 13661 genes common to 5 platforms were filtered for expression variance across 850 samples taken from 38 blood data sets. 3700 genes with low platform variance were subsequently used in a PCA to visualise the behaviour of samples relative to the platform or study that they were sampled from. The outcome of the these steps are shown in Fig 2, where each point represents one sample and the plots show cohesive grouping of similar cell types drawn from different platforms and independent studies. Supplementary S3 Fig shows that most genes retained in the atlas explain a high proportion of variance related to either sample source, progenitor type or cell type compared to platform.

Fig 3 shows the process of implementing progressively stricter thresholds to generate the PCA. Beginning with a permissive threshold of 0.8 platform contribution to total variance in panel **A**, the separation of platforms in the PCA space is clearly evident. As the threshold is lowered to 0.6, 0.4 and 0.2 (panels **B**, **C** and **D**), samples from the different platforms mix and form new clusters. This in turn impacts on the number of genes available to construct the PCA, illustrated in S1B Fig. A threshold of 0.5 leaves approximately 10000 genes, a threshold of 0.25 approximately 5000 genes, and when the threshold is as strict as 0.05, only 500 genes remain.

It is important to note that the figures in S1 Fig are a function of the biology of the samples, combined with the systematic effect of the platform upon genes. As when generating a regular PCA, the variance is affected by the sampling of the underlying biology. If that biology is not well represented, variance modelling will be more difficult. In order to undercover more fine grain detail it is necessary to refine the set of samples and recursively apply this process.

At a global level, the PCA shows clear separation of progenitor types, lymphocytes, and myeloid lineages. The uniformity and stability of these sample groups was confirmed by K-means cluster analysis (see S2 and S3 Tables). S3 Table shows results of the stability analysis performed over a range of $k$ for the K-Means and Agglomerative algorithms. The most stable $k$, as measured by the median of the H-index of the clusters, is highlighted in yellow. In the top right hand column of 4 shows the most stable clustering on all of the blood (including myeloid, lymphocyte and progenitors) is run with $k = 6$. The annotated cell identities in S2 Table show that cluster 1, in the bottom corner, contains the progenitors, Cluster 2 captures lymphocytes and contains the majority of B, T and NK cells. The myeloid lineage is split over a three distinct clusters: Cluster 4 containing circulating monocytes and granulocytes, Cluster 5 predominantly cultured monocytes and tissue-resident macrophages, and Cluster 6 containing dendritic cells.

The large number of different myeloid cell types drives a resolution favouring these subsets. It follows that the resolution of biologically interesting subtypes requires representation from several data sets, and may not resolve if the major biological signal is driven by cell classes that are disproportionately represented. We address this using recursive application of the method,

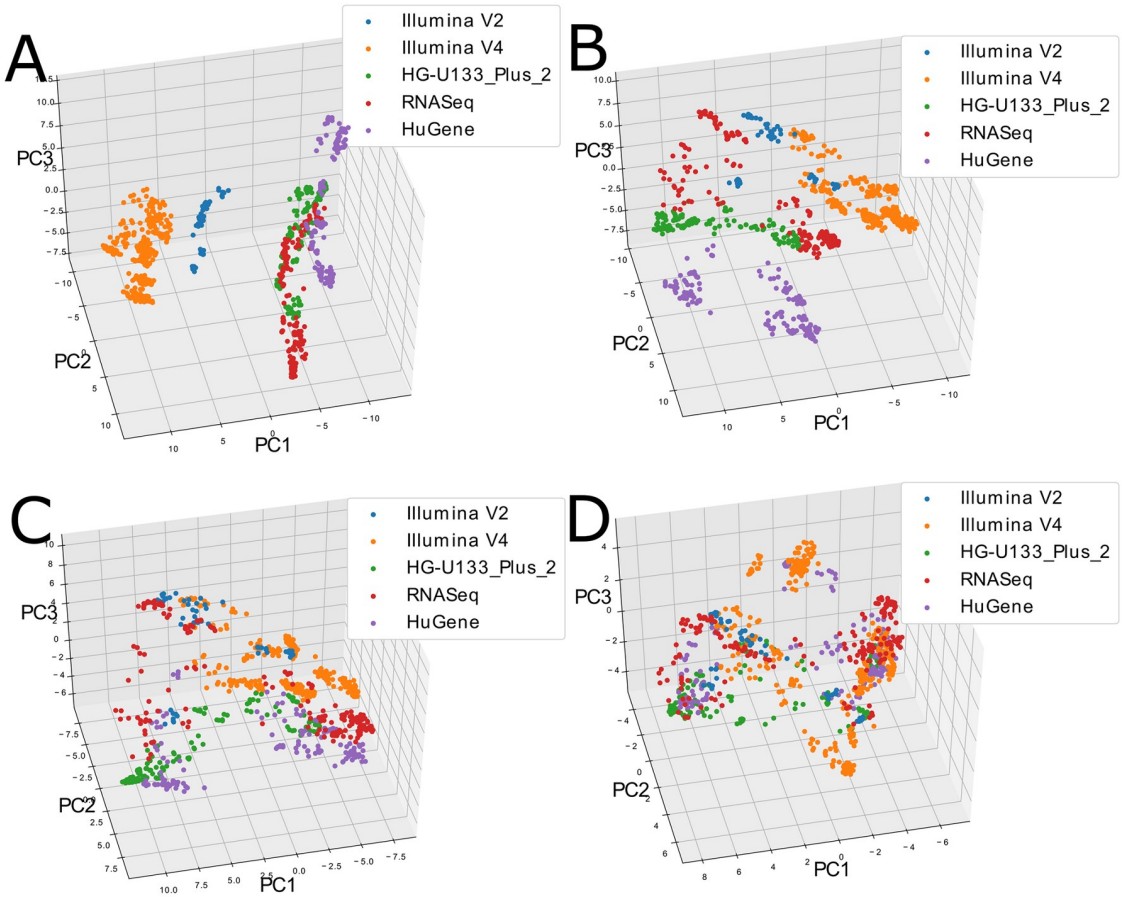

**Fig 3. The PCA coordinates of blood atlas samples after filtering genes with a decreasing platform variance fraction threshold.**
In panel **A** the threshold is 0.8, in **B** it is 0.6, **C** it is 0.4, and **D** is 0.2. As the threshold is lowered, clusters of samples initially separated by platform, merge and form new clusters independent of platform.

on subsets of samples captured in specific regions of the original graph. This allows for ever finer detail and identification of nuanced cell phenotypes, with the limiting factor being the availability of enough data for the biological subset of interest. By using a recursive approach, we view the atlas as a series of blood hierarchies, starting with the most broad categorisation, and moving through smaller sample groupings to find more detailed cell types. For example, in order to resolve the lymphocytes better, the lymphocytes and myeloid arms were isolated from each other and the technique repeated on each. These separate graphs are shown in Fig 4. The PCA on the 728 myeloid samples is performed from the clusters containing progenitors, circulating monocytes and granulocytes, and dendritic cells (and approximately 3600 genes). We further observed differences between circulating and cultured monocytes, naive or activated states, and distinctions between primary or in vitro derived cells (Fig 4 Myeloid). In contrast the PCA on the lymphocytes only include 255 samples from clusters containing lymphocytes and progenitors. The lower number of samples makes it more difficult to resolve structure and results in only approximately 2400 genes being included. Despite the lower number of samples, Fig 4 Lymphocyte shows evident separation of the T and B Cells along the z-axis, as their difference is now strong enough to exceed the platform effect in our gene filter step, however the atlas lacks sufficient samples describing B-cell maturation or identifying phenotypically distinct T-cell classes. At each iteration, a robust global clustering is found for that

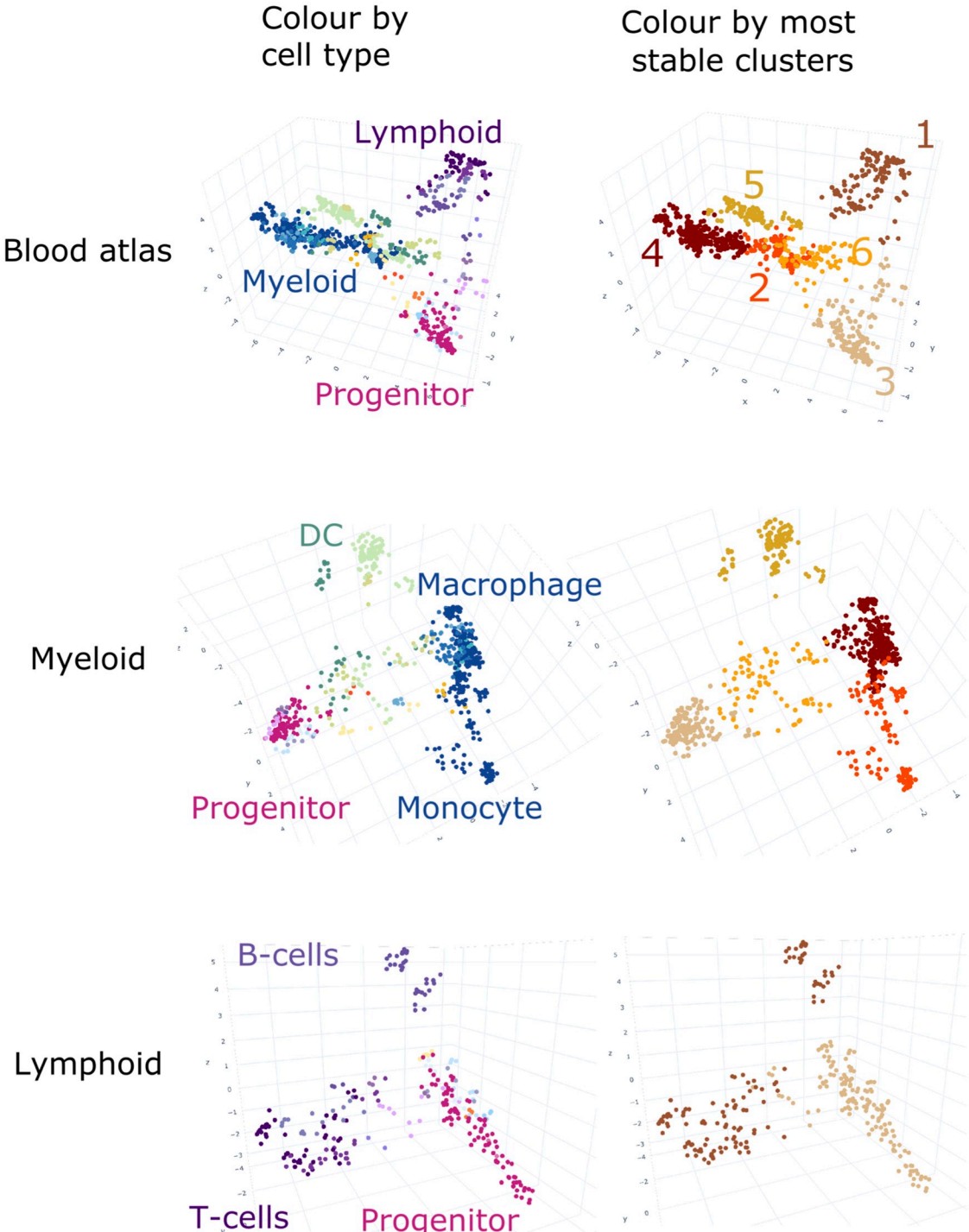

**Fig 4. Repeated application of the gene filtering and PCA upon annotated blood samples.** These panels show the results of repeated application of gene filtering and PCA upon the annotated blood samples in S4M. Each point is a sample, with colour indicative of annotated cell type (left column), or cluster identity (right column). The top left shows application to all blood samples as in Fig 2, while the top right shows the robust clusters defined upon these coordinates. The identities of these clusters are in Supplementary S2 Table. Their highlighted colours are propagated to the middle and bottom panels to display the behaviour of these clusters subsequent to recursive application. The middle row shows the PCA when variance modelling and filtering is applied only to the myeloid lineage clusters (2,3,4,5 and 6). The myeloid PCA shows the clusters defining monocyte, macrophages and dendritic cells separating into distinct regions. The bottom row shows the variance modelling, filtering and PCA upon the lymphoid lineage clusters (1 and 3). Now the increased resolution splits the lymphocyte cluster, 1, into more detailed subsets containing either T or B cells.

scale, and only that scale. By stitching these together, the true multiscale nature of the myeloid arm of the blood hierarchy emerges, and more molecular detail is revealed. The two examples provided here can be further explored in the blood atlas (https://www.stemformatics.org/atlas/blood) and the myeloid subsets in the myeloid atlas (https://www.stemformatics.org/atlas/myeloid) [27].

## Stability

While the biological grouping in the atlas are visually compelling, in order to formally test the stability of the results we ran two types of clustering algorithm on the merged data—the K-Means algorithm and the Agglomerative (bottom-up hierarchical) algorithm. For each approach, we perturbed the underlying data with two re-sampling schemes and measured the stability of cluster membership using the H-index of the Jaccard similarity coefficient (e.g. [28]). Both algorithms, K-Means and Agglomerative, require the number of clusters as an input parameter, and as this number is not known a priori, we test multiple values. The Supplementary S3 and S4 Tables show the results of the jackknife and bootstrap resampling carried out upon clusters found in the respective atlas of each row in Fig 4. These tables list the median ± the maximum and minimum of the H-Index calculated on all clusters after re-sampling. The results of jackknife and bootstrap resampling are qualitatively very similar.

Our clustering defines biological groups by assigning class membership to samples based on their proximity in PCA coordinates. This is our preferred measure of structure because the principal components can (and may be expected to) change under random re-sampling. Groupings of samples ought to be preserved, regardless of coordinate system, if indeed the biological signal is stable and the relative proximities are conserved. If the clustering structure is genuine, stability can be expected up until the point where too many clusters are demanded, after which clusters will be artificially grouped and unstable. We can also expect that the different algorithms should produce similar results if our atlas is stable.

In S3 Table top row, algorithms with cluster numbers up to 6 performed the best. For the both the K-Means and Agglomerative algorithms, the median values in this range are about ∼0.9. This indicates that when re-sampling, the overall structure of the atlas is well preserved. Results for re-sampling the myeloid and lymphoid arms are shown in the middle and bottom rows of S3 Table. The myeloid atlas is stable up until having approximately 5 clusters, at which point they have a high median H-indices of ∼0.9. The lymphocyte atlas is most stable with 4 clusters, but only has median H-index of 0.79 (K-Means) and 0.75 (Agglomerative), and is less stable than other graphs for all of the cluster numbers. This reflects the relatively smaller representation of lymphocyte samples within our data.

We also evaluated the variation of the set of selected genes under re-sampling for the atlas containing all of the blood. Over the 500 bootstrapped iterations the median percentage of genes in common with the true data is 93%, with a minimum of 86% and maximum of %96. For the leave-one-out re-samplings the median similarity is 97%, with a maximum of 99% and minimum of 88%. These indicate the set of genes used to generate the Atlas is also stable to perturbations.

## External data can be projected onto the atlas

Allowing researchers to compare cell types is an important use-case for any robust transcriptional atlas which serves as a reference. This could be to validate or benchmark samples against the reference, to hypothesise about new cell types, or to find key regulators of differentiation. We have taken a simple approach of linearly projecting new data points onto the PCA space of the Blood Atlas for this type of comparison—as we allow users of the website to project their

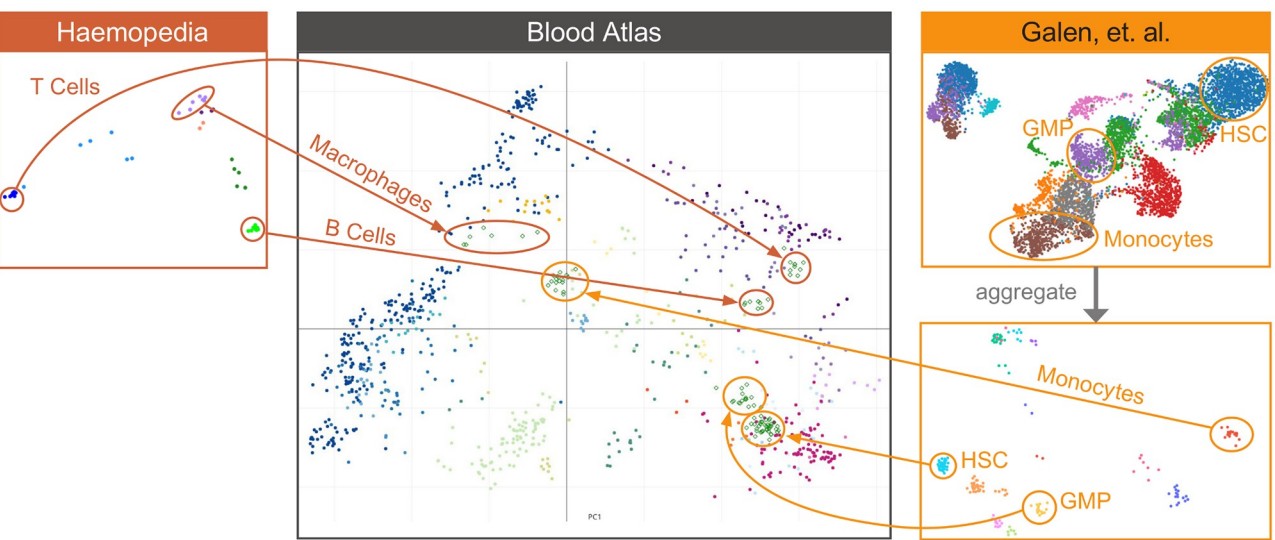

**Fig 5. New and independent samples may be projected onto the Blood Atlas.** The projection of blood data from two external datasets, Haemosphere (https://www.haemosphere.org/) and van Galen scRNA-Seq [26]. The location of the projected data is consistent with the Blood Atlas.

own data this way, a simple approach that users can easily understand is advantageous. We tested projections with a range of different datasets and data types that contain blood samples, to see if they produced expected results. The projections with bulk RNAseq datasets were highly reproducible for both myeloid and lymphocyte arms (Fig 5, [29]). For scRNA-Seq, simply projecting individual cells did not work very well, due to the fundamental difference in the distribution of values in the scRNAseq data compared to platforms. However, we have found that aggregating the samples to simulate pseudo-bulk samples did work well, as shown by Fig 5, [26]. To help users with aggregation and projections in general, there is a vignette which is accessible from the web page. This includes a step-by-step description of the process with examples, including how to decide on parameters of aggregation for scRNAseq data projections.

For samples which are transcriptionally very different to blood cells, such as mesenchymal stromal cells, fibroblasts or neurons, projected coordinates are in the central region of the PCA (S4 Fig). This region corresponds to coordinates where samples sit far away from all regions of the PCA. The web page contains some information about projections for the users, including a caution about interpreting a projection in this region, as well as information about the formats of files to use.

## Discussion

### Gene filtering is an alternative to supervised normalisation

Transcriptional profiling was once a discovery platform used to find new molecules in well-established experimental systems [30]. It is also commonly used to assess cell composition of tissues, or benchmark new cell models by virtue of shared molecular patterns [31]. Typically, researchers will "borrow" samples from other data sets, for example to benchmark or compare to their own experimental system to a previously published standard. Potential biases are introduced by the choice of reference, and this is further compounded by batch correction methods that require the analyst to make prior assumptions about the appropriate categorisation of samples. Methods that require prior determination of biological class force analysts to

make a call about which variables are most important to promote or subtract, or how many biological classes are expected in the merged group. This may be desirable under some circumstances, but arguably less desirable when large data series are compiled, particularly if the normalisation approach inadvertently suppresses important variation across that data series. The addition of new data may require renormalisation of the entire series, limiting the number of comparisons. Without a standardised resource each study's comparator is different to the next, yet such approaches are expected to test the reproduciblity of individual studies. Here we show that the projection of new data onto a reference transcriptome atlas offers a straight forward solution.

In the example described here, the blood cell hierarchy, we demonstrate that when combining a large number of microarray and RNAseq data sets, a basic transformation and gene filter step is all that is required to extract prominent biological features. Supervised batch normalisation methods are very useful when applied to samples with well described properties, and when the split between sample class and technical batch is well balanced. Too often, however, batch and biology is confounded (reviewed in [32]). Supervised normalisation seeks to rescue as many expression points (genes or probes) as possible, so applies a weighted adjustment across the entire gene set. Here we demonstrate that across dozens of data sets, representing hundreds of samples, the variability in gene expression attributable to platform profoundly impacts some, but not all genes. Therefore a weighted adjustment of expression where little prior batch effect is present has the potential to obscure genuine biology. Our approach does not seek to retain all expression measurements but rather constructs the atlas graph only with those expression values that escape a strong batch influence. This is achieved by taking many independent data sets, with minimal processing, to allow the dominant technical, experimental or biological trends to emerge from the combined data series. The resulting blood atlas demonstrably groups cells with common phenotypic attributes in an unbiased manner, and at several scales of resolution of cell type. Minimal processing also easily lends itself to an unsupervised method, which helps prevent over-fitting of sample classes or the biases associated with a restricted reference set.

An alternative method that is gaining attention in the integration of single cell datasets is canonical correlation analysis (CCA) [33]. This method similarly uses a reduced feature selection approach, using only those variables (e.g. genes) that share a linear correlation structure across several data sets, to combine pairs of different experiments into an integrated series. CCA works best when there are a large number of data points in common between the samples to be combined. In contrast, we are combining many datasets of small sample size, such that any pair of individual datasets may lack overlapping cell classes, and in practise often are focused on one particular cell type, such as the humanised mouse models assessing tissue residency of dendritic cells [34] or an in-depth exploration of natural killer cell progenitors in fetal and adult tissues [35]. The correlation between genes and cell types is subsequently explored using PCA.

The question of what is an appropriate normalisation must be assessed in light of the analysis question to be conducted. While there is clearly no 'one-size-fits-all' approach, we acknowledge that there are some limitations to our approach. Simplifying data on a ranked scale removes information about the scale of difference between two points. Consequently some information on gene-gene correlations is lost, although we do allow genes with the same value to keep the same rank. It is apparent from Supplementary Method S1.3 that when combining data from platforms with different expression distributions, the benefits of performing the rank percentile transform outweighs the cons. By applying a gene filtering method, some biologically relevant genes will be removed from the analysis, and this may make it harder for a user to assess sample classes using a marker-based approach. By selecting genes with a low

fraction of variance due to platform, we may we may lose resolution between some biological classes (insofar as variance indicates biological informativeness).

Nevertheless we see in Fig 2 that enough information remains in order to extract a good deal of biological structure, and to find meaningful genes that are driving sample clustering. We also acknowledge that using PCA to review sample behaviour does not allow for examination of non linear relationships between genes or samples. The advantages of rescaling and recursive filtering are clearly demonstrated here, and the resulting expression matrix would be suitable for other graphing or clustering approaches.

## Recursive application to reveal fine-grained or coarse-grained atlas resolution

Blood is arguably the most accessible, and therefore the most comprehensively studied human tissue. The earliest attempts at finding unbiased molecular markers for different cell types came from the "Cluster of Differentiation" (CD) leukocyte markers [36]. In a community effort analogous to the atlas activities today, discovery of CD markers required over 50 laboratories undertaking an antibody screen against panels of blood cells without knowing what the antigen expressed by the cell is, nor what it does—the markers were adopted if they were able to reliably partition different cell types. CD markers are still used today—for example CD14 is a classical marker of monocytes and macrophages [37]; CD4 and CD8 [38, 39] have been adopted into the naming convention of T-cell subsets. Nevertheless, very few of these markers are restricted to one cell type [40], and more typically combinations of markers are required to categorise leukocytes.

Several atlas approaches were proposed to identify molecular markers of blood cell subtypes—these include the microarray profiles in Haematlas from the Bloodomics consortium [41] and Haemopedia which compares RNAseq profiles between mouse and man [29]. While useful, most focus on profiling a small number of cell types in a large number of donors (e.g. QTL studies of monocyte gene expression) or a large number of well characterised cell types in a small number of donors [11, 42]. However direct comparison between these projects is very difficult because of discrepancies in the way data is captured, and this is the problem addressed by the integrated atlas approach proposed here.

When considering what are the prominent biological features of any collection of data it is important to remember that 'prominent' is relative. The difference between lymphocytes and myeloid cells may be prominent when looking at blood cells only, but when compared to stem or stromal cells, it could be reasonably said that lymphoid and myeloid cells look very similar. The recursive approach is crucial to our analysis—at each level, the dominant global structure is retrieved and used to inform the next iteration, thus avoiding to impose global axis on all cells which may not reflect small scale structure. Therefore, the recursive approach is an intuitive way to map the cell landscape.

That this multi-scaled nature of a biological system can be captured recursively provides us with new opportunities to review phenotypes that might be expected to deviate from a reference atlas. Examples of future atlases might include disease states that fundamentally alter cell state, or experimental manipulation that creates new cell types. For these applications we recommend a multi-tiered approach, first projecting the new cell types to the current reference atlas to assess similarity to the groups included in the reference, then recompiling the atlas with the disease samples included to allow for additional biological variance specific to the disease to be captured in the new atlas. A leukemia atlas, for example, would comprise both healthy and leukemic cell types. An inflammatory atlas would include naive and activated cell

types, and so on. The critical consideration here is adequate replication across data sets of the sample categories that are included in any rederived atlas.

This highlights a second important limitation to our approach: the necessity for large amounts of diverse data, covering different cell types and experimental platforms. Subsampling regions of the atlas and applying a new round of gene filtering is a recursive approach that allows users to scale between global (all samples) or local cell comparisons. This extracts the most dominant structure at each resolution level, however with fewer samples we also approach the limits of our technique, and the results may become less robust. This can be observed in the lymphocyte arm of the atlas, which in the current iteration are represented by only a few data sets (Fig 2). The resolution of these cells is adequate at lineage level (B-cells vs T-cells) but with only 255 samples, it does not resolve subtypes of T Cells, such as CD4 or CD8. In contrast, resolution between different myeloid subsets is very high, and the emergent properties of the myeloid atlas highlight the impact of experimental handling or derivation method on the type of macrophage or DC studied.

## Data projections and integration of single cell platforms

Given the advent and popularity now of single cell sequencing, future iterations will see the inclusion of single cell data. Deeper molecular characterisation of individual cells could be expected to better resolve functionally discrete populations, as well as provide new candidate markers for prospective cell isolation and characterisation. With the blood atlas method, we aim to provide a reference benchmark that evaluates past transcriptomic data through a novel and relatively simple integration approach, and use this for comparisons to new data types, including scRNAseq of blood cells from different tissues. In the current iteration, we show the usefulness of projection of scRNA data onto the atlas, particularly for identification of blood cell types and annotation of scRNAseq clusters.

While we use the graph space obtained by the combined atlas series to project new data into the predefined state space, it's important to note that we are not using this to 'tune' new data sets into this space. Other graph smoothing methods have been described [10, 43], and particularly applied to the integration of single cell batches, where 'harmonisation' of the combined data is achieved by iterative weighting of gene expression in the introduced samples. Here, data set projections are used first to the reproduciblity of cell groups and group annotations using external, independent data. Secondly, projections of single cell expression data into prior annotated groups is used to lift atlas annotations over to the single cell experiment.

Projection strategies may provide additional benefits. Since we rank transform each sample before projection, each sample is treated independently to assess its similarity to the atlas cell types. Hence insights can be gained from data sets where batch effects are already confounding interpretation of data in the original experimental series—for example, where each sample class is obtained in a separate technical batch. In this instance, projection of each set of samples onto a reference atlas allows for examination of the experimental groups against an unbiased set of relevant cell classes. Projections may also inform trajectory analyses for scRNAseq datasets, without having to derive these trajectories de novo. For example, plotting single cell clusters from a differentiation series onto the blood atlas will allow better identification of haematopoietic cell lineages, or even suggest new pathways of differentiation, especially in cases where scRNAseq data come from cell types with low coverage within the Blood Atlas.

## Conclusion

A shift from data collection on successive technologies, to integrated analyses across series of data offers an opportunity to view biological collections across a hierarchy of perspectives and

information. In the example given here, we recapitulate the haemopoietic systems by combining 38 datasets, each describing detailed aspect of one part of that system in a small number of donors. The result is a multi-scaled tool to visualise and analyse the transcriptional relationships in the blood cell lineage. This allows users to rapidly review gene expression across a large number of samples to find reproducible markers of cell type, and new markers correlated to derivation method, culture condition, or extrinsic signals.

Recursive application of the method was demonstrated by the general categorisation seen in whole blood to the identification of specific myeloid cell types and activation states in the accompanying myeloid atlas. The projection of additional data onto the atlas, provides a tool for researchers to compare their own data to a robust reference collection. Projection of single cell data provides definitive annotations of blood cell clusters without prior assignment of marker genes in the scRNA-seq data.

The method is simple and scalable, so providing anyone with the means to curate their own reference atlas to address additional biological systems. Implementation of the blood and myeloid atlases provides a simple web-based tool in the Stemformatics platform.

## Supporting information

**S1 Text. Supplementary method.** Expanded description of the gene filtering, rank transformation and stability analysis.
(PDF)

**S1 Fig. Statistics of the fraction of variance attributable to platform. A**: The distribution of the fraction of variance attributable to platform for the blood data. It is weighted towards low ratios, indicating that biological variation forms a major part of the signal. **B**:The total number of genes that pass the cut for platform variance thresholds from 1.0 to 0.
(TIFF)

**S2 Fig. The change in value of the Kruskal-Wallis H Test for the first 10 components of the Blood PCA as the platform-variance threshold is decreased.** Lower values for the KWH test indicate platform has less effect upon the component. Lowering the threshold has the effect of both moving platform-related components to lower components, and decreasing the overall dependence upon platform.
(TIFF)

**S3 Fig. Proportion of variance explained by platform, residuals and either A: Sample Source, B: Progenitor type or C: Cell type assessed with a linear mixed model.** Each gene is depicted as a vertical line on the x-axis, and genes are ranked according to the ratio Sample Source / Platform explained variance. Dark gray vertical lines indicate genes that were retained in the filtered data set.
(TIFF)

**S4 Fig. The projection of example non-blood data (induced pluripotent stem cells, mesenchymal stem cells, fibroblasts, neurons) onto the Atlas.** They are displayed as green crosses. They sit in a region low on component 2, a region not populated by either by the blood samples used to generate the atlas.
(TIFF)

**S5 Fig. Comparisons of batch effect correction approaches on the simulated study described in supplementary method S1.3 using t-SNE.** Ten cell types are indicated by colors. **(A)**: original count data include a batch effect across 4 platforms. **(B)**: correction for platform

effect with limma followed by voom transformation, **(C)**: Combat and **(D)**: percentile rank transformation.
(TIFF)

**S1 Table. Sample metadata.** Table containing metadata for each of the samples used in construction of the Blood Atlas PCA. Also lists the datasets used to project external samples onto the PCA.
(XLS)

**S2 Table. Tables containing the number of the samples that belong to each cluster in the K-Means clustering analysis.**
(DOCX)

**S3 Table. Results of the jackknife resampling stability analysis.** Most stable number of clusters, the median H index, and their maximum/minimum H index as the superscript/subscript.
(XLSX)

**S4 Table. Results of the bootstrap resampling stability analysis.** Most stable number of clusters, the median H index, and their maximum/minimum H index as the superscript/subscript.
(XLSX)

## Acknowledgments

The authors thank Matthew Rutar for early discussions on blood annotations, Jack Bransfield and Isha Nagpal for front end development of the Stemformatics server. This research was undertaken with the assistance of resources from the National Computational Infrastructure (NCI Australia), an NCRIS enabled capability supported by the Australian Government.

## Author Contributions

**Conceptualization:** Paul W. Angel, Christine A. Wells.

**Data curation:** Nadia Rajab, Chris M. Pacheco, Tyrone Chen, Christine A. Wells.

**Formal analysis:** Paul W. Angel, Yidi Deng, Tyrone Chen, Jarny Choi.

**Funding acquisition:** Nadia Rajab, Kim-Anh Lê Cao, Christine A. Wells.

**Investigation:** Paul W. Angel, Yidi Deng, Jarny Choi.

**Methodology:** Paul W. Angel.

**Project administration:** Jarny Choi, Christine A. Wells.

**Software:** Paul W. Angel, Yidi Deng, Jarny Choi.

**Supervision:** Kim-Anh Lê Cao, Jarny Choi, Christine A. Wells.

**Validation:** Paul W. Angel, Yidi Deng, Jarny Choi.

**Visualization:** Paul W. Angel, Yidi Deng, Kim-Anh Lê Cao, Jarny Choi.

**Writing – original draft:** Paul W. Angel, Christine A. Wells.

**Writing – review & editing:** Paul W. Angel, Nadia Rajab, Tyrone Chen, Kim-Anh Lê Cao, Jarny Choi, Christine A. Wells.

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
