## [Decision Letter · Decision Letter 0]

26 Jun 2020

Dear Professor Wells,

Thank you very much for submitting your manuscript "A simple, scalable approach to building a cross-platform transcriptome atlas" for consideration at PLOS Computational Biology.

As with all papers reviewed by the journal, your manuscript was reviewed by members of the editorial board and by several independent reviewers. In light of the reviews (below this email), we would like to invite the resubmission of a significantly-revised version that takes into account the reviewers' comments.

We cannot make any decision about publication until we have seen the revised manuscript and your response to the reviewers' comments. Your revised manuscript is also likely to be sent to reviewers for further evaluation.

Sincerely,

Elana J. Fertig, PhD

Associate Editor

PLOS Computational Biology

Erik van Nimwegen

Deputy Editor

PLOS Computational Biology

Reviewer's Responses to Questions

**Comments to the Authors:**

Reviewer #1: Thanks for you inviting me to review this paper.

It is an excellent integrative analysis tool which should enable the re-use of a significant quantity of the transcriptomic data that has already been generated via older methods such as micro-arrays and bulk RNAseq and also provides possibilities for integrative analysis in the scRNAseq space

Your paper explains your method clearly explaining both its advantages compared to other approaches and its drawbacks.

You have made the methods and the data all openly available via the https://www.stemformatics.org/ and code sharing repos

I have some small readability improvements that I would like to suggest, please re-use these as you see fit

Page 3/24 - Line 9

I think

"Yet technology change drives much recursive data derivation"

would read better as

"Yet technological change drives much recursive data derivation"

page 3/24 - Line 12

"catalogues" should be "catalogue"

page 5/24 - Line 23

"Gene expression for each samples"

should this be

"Gene expression for each sample"

page 5/24 - Line 24

"the highest expression gene is assigned a value 1 and the lowest receives a value 0"

would read better

"the gene with the highest expression is assigned a value of 1 and the lowest receives a value of 0"

page 18/24 - Line 25

"by only a few data sets 2"

should this be

"by only a few data sets"

Thank you for providing this excellent analysis and software to the community. I look forward to seeing it more broadly used to support single cell analysis.

Laura Clarke

Reviewer #2: Angel et al. present a resource that combines a large number of hematopoietic cell profiling datasets through a simple and effective normalization method, offered through an online portal and open source code. This could be a valuable resource, and while is currently limited in scope to hematopoiesis, is a great proof of principle of combining disparate datasets for biological analysis. I have several questions or concerns about the implementation of this atlas that are listed below.

Major points

1. Platform effect. I appreciate the method the authors use to deconvolve platform effect - it is intuitive, and it also seems sensible to use subsets of genes, since using very large gene sets (tens of thousands) for analyses tends to decrease the signal-to-noise. However in the main manuscript there is little emphasis on the platform effect and the impact of the choice of gene set. I would suggest moving some SI figs to main? E.g. Fig S4. It would also be helpful to have more detail about the impact of the size of the input gene set on the PCA (as in S4 or similar) - what variance threshold corresponds to 500 or 5000 genes, etc? Also: if datasets between platforms differ by *biological* variation, e.g. different (proportions of) cell types, how does the variance threshold handle this?

2. PCA. Why were 10 principal components chosen? Is there justification for their use or any comparison with other possible choices?

3. Details of biological datasets included are lacking, e.g. disease condition? I notice that some datasets describe pathologies (e.g. PMID 27630125, CML): would pathological myeloid cell populations not be expected to lie far from healthy hematopoietic populations? Is there a way to tag disease states and other features on the atlas?

4. In more general terms, while I do see the value of this resource, what are the use-cases that the authors imagine? E.g. biological discovery? In which case one would need additional labeling of the samples, by cell type or disease state.

5. It is unclear how single-cell datasets are handled. Please expand on the “pseudo-bulk” aggregation is performed. I cannot find Galen et al. in the list of datasets. (Also general problem even for datasets listed: I can’t find their placement on the plot.) Moreover, the projection of new single-cell datasets onto this atlas is likely to be a popular task, given the amount of interest in this area, so other examples would be extremely helpful. What is minimum “sample” size (i.e. # of aggregated single cells)? Could single cells be rank transformed and projected? More detail and examples here would greatly expand the utility of this tool.

6. Blood is special, in its level of cell type characterization, ease of sampling, and availability of cell surface markers to sort for particular populations. Can similar methods be used to create similar integrated datasets for other tissues?

Minor points

- “iMAC” is probably not a good choice of name for one atlas - I imagine Apple would take issue with this if the tool become popular enough, I would advise changing it

- not clear online how to find a dataset on the plot, e.g. If I click “find dataset” then choose one, it does not show up on the plot?

- Fig 1 needs to be updated to be more useful than obfuscating: red circles are identical across platforms, this is confusing; vertical arrows have no annotation; top right gray box does not really add information

**Have all data underlying the figures and results presented in the manuscript been provided?**

Reviewer #1: Yes

Reviewer #2: Yes

PLOS authors have the option to publish the peer review history of their article (what does this mean?). If published, this will include your full peer review and any attached files.

Reviewer #1: **Yes: **Laura Clarke

Reviewer #2: No
---

## [Decision Letter · Decision Letter 1]

4 Aug 2020

Dear Professor Wells,

We are pleased to inform you that your manuscript 'A simple, scalable approach to building a cross-platform transcriptome atlas' has been provisionally accepted for publication in PLOS Computational Biology.

Best regards,

Elana J. Fertig, PhD

Associate Editor

PLOS Computational Biology

Erik van Nimwegen

Deputy Editor

PLOS Computational Biology

Reviewer's Responses to Questions

**Comments to the Authors:**

Reviewer #1: Thanks for addressing both mine and the other reviews comments

Reviewer #2: The changes made in this revision have greatly improved the clarity of the paper and the presentation of results. All of my questions have been addressed. I have one further comment, following on from a point the authors raise in their response letter.

Regarding the future inclusion of perturbed states (e.g. disease or inflammation) in an atlas, the authors state that “The critical consideration here is adequate replication across data sets of the sample categories that are included in any rederived atlas.”

In addition to the iterative design that will be required to characterize disease states, the increase in variability often observed in disease (which some have termed the “Anna Karenina effect,” see the first sentence of the book) likely compounds the challenge of adequate replication in the case of disease. I think these additional challenges are important to note.

**Have all data underlying the figures and results presented in the manuscript been provided?**

Reviewer #1: Yes

Reviewer #2: Yes

PLOS authors have the option to publish the peer review history of their article (what does this mean?). If published, this will include your full peer review and any attached files.

Reviewer #1: No

Reviewer #2: No

---

## [Editor Report · Acceptance letter]

18 Sep 2020

PCOMPBIOL-D-20-00503R1 

A simple, scalable approach to building a cross-platform transcriptome atlas

Dear Dr Wells,

I am pleased to inform you that your manuscript has been formally accepted for publication in PLOS Computational Biology. Your manuscript is now with our production department and you will be notified of the publication date in due course.

With kind regards,

Laura Mallard
